# MagC, magnetic collection of ultrathin sections for volumetric correlative light and electron microscopy

Thomas Templier[†]*

Institute of Neuroinformatics, University of Zurich and ETH Zurich, Zurich, Switzerland

**Abstract** The non-destructive collection of ultrathin sections on silicon wafers for post-embedding staining and volumetric correlative light and electron microscopy traditionally requires exquisite manual skills and is tedious and unreliable. In MagC introduced here, sample blocks are augmented with a magnetic resin enabling the remote actuation and collection of hundreds of sections on wafer. MagC allowed the correlative visualization of neuroanatomical tracers within their ultrastructural volumetric electron microscopy context.
DOI: https://doi.org/10.7554/eLife.45696.001

*For correspondence:
thomas.templier2@gmail.com

Present address: [†]Ecole Polytechnique Fédérale de Lausanne, Lausanne, Switzerland

## Introduction

The ultrathin physical ablation of sample blocks is a prerequisite for volumetric biological electron microscopy (EM). The destructive methods, serial block face (*Denk and Horstmann, 2004*) and focused ion beam EM (*Knott et al., 2008*), enable serial access to the sample in its whole depth only very briefly and inside the vacuum chamber of a specialized scanning EM, prohibiting the (re-) imaging of permanently destructed portions, liquid treatments such as heavy-metal poststaining or immunostaining (*Micheva and Smith, 2007*), fluorescent light microscopy (LM) (*Sigal et al., 2015*), and various nanoscale imaging techniques (*Pirozzi et al., 2018*).

The automated non-destructive tape-based ablation method ATUM (*Kasthuri et al., 2015*), which has greatly benefited volumetric EM (*Kornfeld and Denk, 2018*), provides sections on silicon wafers but at a low packing density (about 200 per 100 mm diameter wafer), through an intermediate tape, and after manual gluing onto a wafer. Carbon-coated Kapton tape suffers from strong autofluorescence that prevents fluorescence microscopy, from scratches that impair EM imaging, and from the problem that it is not easily carbon-coated uniformly (*Kubota et al., 2018*) (a step that is necessary to avoid charging during imaging). Recently introduced carbon-nanotube tapes (*Kubota et al., 2018*) solve most of these issues, although they require a custom device for reel-to-reel plasma hydrophilization and manual grounding of all cut tape stripes with conductive tape on top. Other non-destructive collection approaches require excellent ultramicrotomy skills and intense monitoring of ribbons during sectioning and collection onto flat substrates (*Horstmann et al., 2012*; *Spomer et al., 2015*; *Burel et al., 2018*; *Koike et al., 2017*; *Smith, 2018*; *Templier and Hahnloser, 2019*).

The invention of MagC was motivated by the wish to collect sections directly onto silicon wafers for excellent fluorescent LM and EM imaging conditions without tape-related issues and at high packing density, thereby facilitating convenient bulk staining procedures with liquids and uninterrupted imaging in automated fluorescent light and electron microscopes, including next-generation multibeam electron microscopes (*Eberle and Zeidler, 2018*). In MagC, a piece of resin containing superparamagnetic nanoparticles is glued onto a polymerized sample block so that all cut sections carry magnetic material. Remote magnetic actuation then allows the agglomeration of floating

sections in the center of a large bath attached to a diamond knife until they are deposited onto an underlying silicon wafer. Finally, the order of the sections is retrieved computationally after section collection. Two volumetric correlative LM-EM datasets of connectomics-grade brain tissue are presented here.

## Results

### Magnetic resin

Magnetic epoxy-based resin containing 8% (w/w) iron oxide superparamagnetic nanoparticles (*Puig et al., 2012*) was produced for remote actuation. The resin also contained fluorescent polymer beads for post-collection section order retrieval, and a fluorescent dye to ease section segmentation (*Figure 1a*, *Figure 1—figure supplement 1*, *Figure 1—figure supplement 2*). A piece of this resin was glued with an epoxy that is usually used for EM studies (durcupan) to a sample block of interest, with the help of a small mechanical device (*Figure 1—figure supplement 3*), thereby maintaining the position of the blocks during curing in an oven. The resulting blocks were trimmed, and a second piece of resin (the 'dummy'), consisting of a piece of heavy-metal stained and resin-embedded brain tissue, was glued to the magnetic resin to enhance cutting quality. The final block assembly was trimmed for ultrathin sectioning (*Figure 1a*).

### Sectioning

A custom diamond knife was built with an enlarged bath to let many hundreds of sections float at the water's surface (*Figure 1b*). A hole was drilled in the bottom to fill and empty the bath with a motorized syringe pump. A piece of silicon wafer was immersed in the bath and was slightly

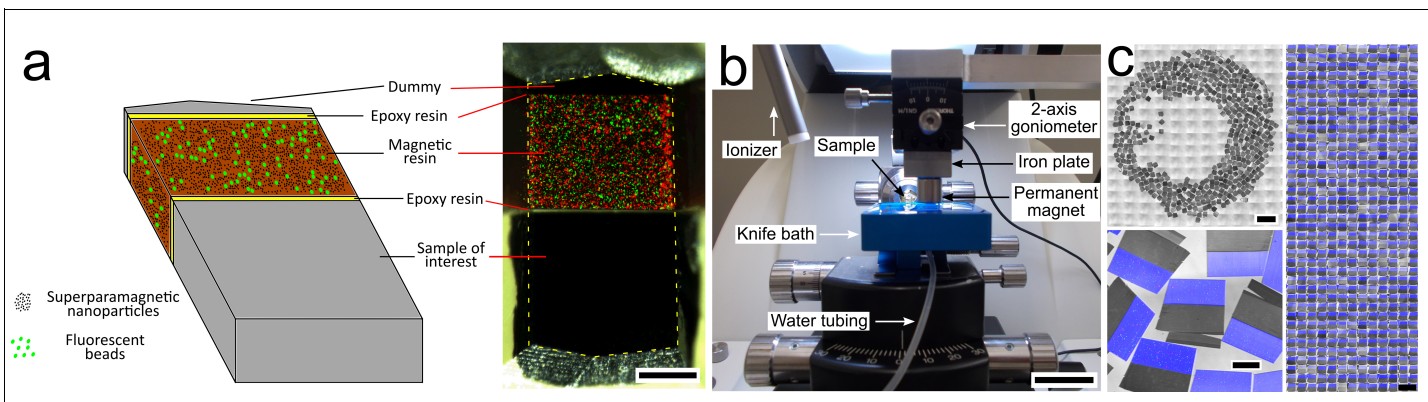

**Figure 1.** Magnetic augmentation and collection of sections on silicon wafer. (**a**) Augmentation of a polymerized sample block with resin containing superparamagnetic nanoparticles (for remote magnetic actuation) and fluorescent beads (for section order retrieval). (**b**) Setup for MagC: a diamond knife with a large bath and a mobile overhanging magnet. (**c**) 507 consecutive ultrathin sections collected on a silicon wafer: wafer overview, close-up (merge of whitefield and three fluorescent channels: blue for coumarin stain, plus green- and red-fluorescent beads) and montage of all sections. Scale bars: (a) 200 μm; (b) 2 cm; (c) 2 mm (top left), 200 μm (bottom left), 1 mm (right).
DOI: https://doi.org/10.7554/eLife.45696.002

The following figure supplements are available for figure 1:

**Figure supplement 1.** Wafer overview for Dataset 2.
DOI: https://doi.org/10.7554/eLife.45696.003

**Figure supplement 2.** Electron micrographs of the sections for Dataset 1.
DOI: https://doi.org/10.7554/eLife.45696.004

**Figure supplement 3.** Magnetic augmentation.
DOI: https://doi.org/10.7554/eLife.45696.005

**Figure supplement 4.** Wafer overview of the 100 sections collected during the video-recorded session.
DOI: https://doi.org/10.7554/eLife.45696.006

**Figure supplement 5.** Multibeam scanning EM of magnetically collected sections on silicon wafer.
DOI: https://doi.org/10.7554/eLife.45696.007

tilted relative to the water level (by about two degrees) to avoid accumulation of water surface dust in the center of the wafer at the end of the water withdrawal. After alignment of the knife and cutting a few sections, automatic sectioning was started and allowed to continue uninterrupted until the last section was cut. A ionizer whose tip was placed close to the diamond knife created a very soft air current that gently detached sections from each other every few sections without impairing the cutting process. The sections floated freely at the water surface.

## Magnetic collection

To collect the floating sections after the sectioning, a permanent magnet (cylindric, 15 mm diameter x 8 mm) was placed above the water surface with a 1 mm air gap (*Figure 1b*). A few sections (about a dozen) that were loosely attached to the walls of the bath were gently detached with an eyelash probe and remained undamaged. The magnet, actuated by a robotic arm, scanned the surface of the water bath, taking a snaking path. At the end of the scan, the sections were accumulated in the center of the bath. Water was then withdrawn with a motorized syringe pump, while the 1 mm air gap was maintained by lowering the magnet with manual robotic control. Two small heating pads placed below the bath were turned on when the water level reached the level of the substrate. The elevated wafer temperature generated by the heating pads (about 40 °C) accelerated the evaporation of the water left at the wafer surface and avoided the formation of wrinkles in the deposited sections. The wafer was finally placed on a hot plate at 50 °C for 30 min. I report here on two wafers of 507 (Dataset 1, *Figure 1c*) and 203 consecutive 50-nm-thick sections (Dataset 2, *Figure 1—figure supplement 1*). The magnetic collection of these datasets was not video recorded, but *Video 1* shows another small complete MagC experiment with 100 consecutive 50 nm thick sections, and *Figure 1—figure supplement 4* shows the sections on the silicon wafer.

## Order retrieval

The serial order was lost during the sectioning and had to be retrieved. After low-resolution (5x) reflection whitefield and fluorescent imaging of the wafers, the location and orientation of the sections was semi-automatically inferred. After calibration of four landmarks, medium resolution (20x) fluorescent imaging was automatically performed on the magnetic portion of each section. The cloud of fluorescent beads (2 μm mean diameter) contained in the magnetic resin was revealed in this imagery. As the section thickness (50 nm) was smaller than the diameter of the beads, each bead was visible in at least a dozen consecutive sections, so that the pairwise similarity of all sections could be computed. Solving a traveling salesman problem on the graph of pairwise similarities retrieved the serial order (which was confirmed later manually with EM) with only a single error occurring when a high concentration of beads (1% w/w) was used (*Figure 2a*). A lower concentration yielded more errors (0.2% w/w, *Figure 2—figure supplement 1*). With the same methodology, the serial order could also be retrieved using the brain tissue EM imagery (*Figure 2a*). Note that order retrieval using fluorescent beads contained in the magnetic resin does not depend on the processed sample, which makes MagC suitable for collecting samples that, for example, would not show sufficient information for order retrieval by LM or EM.

## Imaging

The high packing density of the collected sections on wafer allowed convenient staining

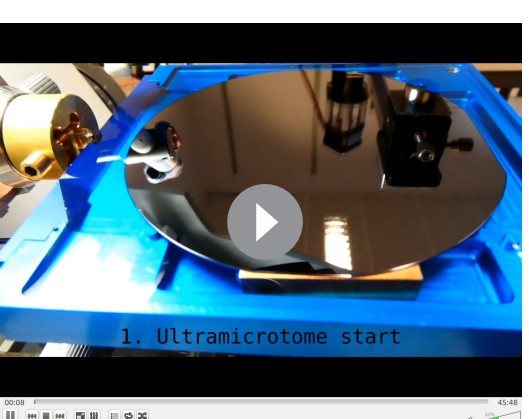

**Video 1.** Video-recorded magnetic collection. Video available here: https://youtu.be/o13r-tHT9-c. Timeline:1. 00:02 - Ultramicrotome start, 2. 00:19 - Cutting ..., 3. 06:21 - Cutting stopped, 4. 06:32 - Removal of ionizer, 5. 07:02 - Magnet scanning ..., 6. 17:25 - Blowing away 2 sections from wall, 7. 20:42 - Blowing away 1 section from wall, 8. 27:15 - Water removal ..., 9. 31:33 - Heating ..., 10. 45:41 - Wafer pickup.

DOI: https://doi.org/10.7554/eLife.45696.017

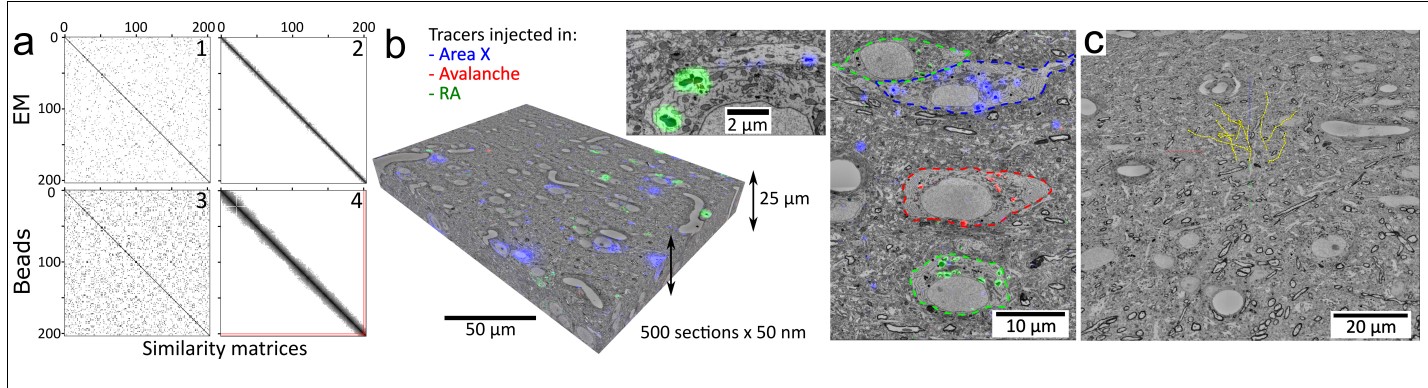

**Figure 2.** Volumetric correlative LM-EM with MagC collected sections.Volumetric correlative LM-EM with MagC collected sections. (a) Section-order retrieval for Dataset 2 (1% fluorescent beads) obtained with EM imagery (panels 1 and 2 show the pairwise similarity matrices before and after reordering, respectively) and with fluorescent beads imagery (panels 3 and 4). Darker pixels depict higher similarity whereas white pixels depict no similarity. The two red lines in panel 4 indicate a single flip in the computed order that was later corrected with EM imagery. (b) Volumetric correlative stack for Dataset 1 with three fluorescent channels and 507 consecutive ultrathin sections. Insets: close-ups of cell bodies and a neurite carrying different neuroanatomical tracers. The cell bodies in the right panel are outlined with colored dashed lines. Blue: tracer injected into Area X. Green: tracer injected into the nucleus Robustus of the arcopallium (RA). Red: tracer injected into Avalanche. (c) The EM imagery was connectomics-grade and enabled neurite tracing. Yellow dots: skeletons stemming from nine seed points placed in a 3×3 grid in the first section of Dataset 2.
DOI: https://doi.org/10.7554/eLife.45696.008

The following figure supplements are available for figure 2:

**Figure supplement 1.** Section order retrieval for Dataset 1 (low concentration of beads).
DOI: https://doi.org/10.7554/eLife.45696.009

**Figure supplement 2.** Automated LM-EM registration.
DOI: https://doi.org/10.7554/eLife.45696.010

**Figure supplement 3.** Sectioning quality and restart of a MagC block.All images are SEM micrographs of sections manually collected on a silicon wafer.
DOI: https://doi.org/10.7554/eLife.45696.011

**Figure supplement 4.** Labeled axon making a synapse en passant.
DOI: https://doi.org/10.7554/eLife.45696.012

**Figure supplement 5.** Section collection quality.
DOI: https://doi.org/10.7554/eLife.45696.013

**Figure supplement 6.** Multicolor correlative LM-EM imagery (Dataset 1) visualized in neuroglancer.
DOI: https://doi.org/10.7554/eLife.45696.014

procedures (simply exchanging a few microliters of staining solution repetitively on an area smaller than 2 cm x 2 cm), and easy loading into LM and EM microscopes for uninterrupted automated imaging. After immunostaining against neuroanatomical tracers previously injected into the brain of two zebra finches, multichannel fluorescent imaging (3 and 1 fluorescent channels in Datasets 1 and 2, respectively, and one widefield channel) was automatically performed with custom scripts. Note that the small wafers were easily coverslipped with mounting medium underneath and oil on top to enable reflection LM with high magnification immersion objectives (63x). After washing off the mounting medium on the wafer followed by heavy metal poststaining, automated scanning EM was performed with custom scripts, acquiring the same portion of each section (*Video 2*). Volumetric EM imagery was assembled (contrast enhancement, stitching, affine then elastic alignment) and the LM modality was registered to its EM counterpart (*Figure 2—figure supplement 2*). The whole processing chain was entirely automated with custom scripts (*Templier, 2019*) (https://github.com/templiert/MagC; copy archived at https://github.com/elifesciences-publications/MagC) operating in the Fiji/TrakEM2 environment (*Schindelin et al., 2012*; *Cardona et al., 2012*). Multibeam scanning EM was also used successfully to image magnetically collected sections (*Figure 1—figure supplement 5*) in two settings: without intermediate treatments after collection (no immunostaining, no mounting, no LM, no poststaining), and after performing the entire CLEM pipeline on the wafer of Dataset 1 (immunostaining, mounting, LM, washing, poststaining, single beam EM) and an additional wafer-wide broad ion beam milling of about 25 nm (Templier, in preparation).

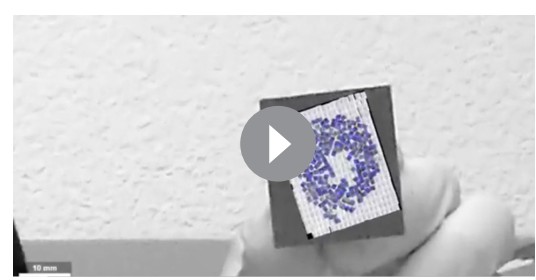

**Video 2.** Zoom on wafer of Dataset 2. Video available here: https://youtu.be/UC8Zrl2Xud4.
DOI: https://doi.org/10.7554/eLife.45696.015

## Data analysis

The experiments yielded correlative LM-EM stacks of brain tissue ready for connectomic analysis (*Figure 2b* and *Video 3*). For convenient use, the data were converted to the neuroglancer format and hosted online for seamless browsing and annotation with the web-based tool neuroglancer, also using the enhancements for multichannel overlay offered by neurodataviz (*Figure 2—figure supplement 6*). To demonstrate the suitability of the data for connectomic analysis, I traced nine neurites with starting points located on a 3 × 3 grid within a central area of the first section (*Figure 2c*). The neurites spanned the whole depth of the dataset across all sections (10 µm) and exhibited a rather axial orientation, amounting to a total length of at least about 90 µm. I also identified structures that were tagged with an injected neuroanatomical tracer such as an axon, making an *en passant* synapse (*Figure 2—figure supplement 4*).

## Assessment of sectioning quality and restart experiment

In Dataset 1, three tears were found in the EM imagery spanning a total of nine sections out of the 507 sections. A first tear was present in a single section #258 (*Figure 2—figure supplement 5c*), then a similar tear was visible in section numbers #480, #481, #482, #483, #484 and #485 (#481 shown in *Figure 2—figure supplement 5e*). The location of the tears looked identical, indicating a potential knife weakness at this location. A third tear was present at a different location in sections #359 and #360 (*Figure 2—figure supplement 5d*). Nevertheless, the EM imagery in the dataset (and especially before and after the tears) did not exhibit the usual vertical streaks caused by knife defects, hinting at a mild temporary knife defect. In Dataset 2, no tears were present in the tissue portion of the sections. Just a single tear was present in the magnetic portion of one section (section #22, *Figure 2—figure supplement 5b*).

In addition, an experiment was performed with a magnetically augmented block (one section shown in *Figure 2—figure supplement 3a*) with two purposes: (i) to assess knife damage due to the magnetic resin by cutting about 5000 sections with the same knife portion and counting the number of sectioning fails between sections #4000 and #5000, and (ii) to assess the continuity of the imagery across a sectioning restart that included a sham MagC collection (see 'Materials and methods') and a 20 hr long break. The timeline of the sectioning experiment is shown on the right side of panel (b) in *Figure 2—figure supplement 3*.

In the high-resolution (8 nm/pixel) imagery acquired from a horizontal band (25 µm x 870 µm, yellow box in *Figure 2—figure supplement 3a*) in sections number #100, #500, #1000, #1500, #2000, #2500, #3000, #3500, #4000, and #5000, a total of three occurrences of vertical streaks (rather mild) were found in sections #1000 and #2000 (shown in *Figure 2—figure supplement 3b and d-1,2,3*). The first streak in section #1000 (*Figure 2—figure supplement 3*, panel d1) was no longer present in section #1500, but a pair of closely spaced vertical streaks appeared again at section #2000, probably at the same location (shown in *Figure 2—figure supplement 3d3*). Both the second vertical streak in *Figure 2—figure supplements 3d2* and the aforementioned streak pair disappeared after the knife cleaning at section #2000. After that, no more vertical streaks were observed and the overall

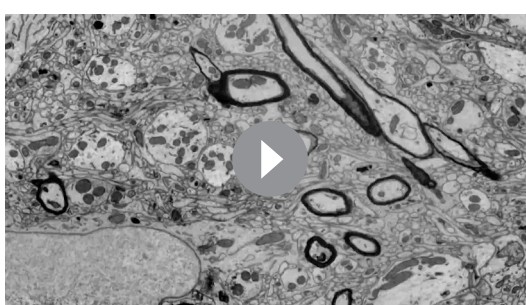

**Video 3.** Flythrough in EM imagery of Dataset 2. Video available here: https://youtu.be/VL0F9DkZVaQ and associated data available at https://neurodata.io/data/templier2019/.
DOI: https://doi.org/10.7554/eLife.45696.016

tissue quality observed was well represented by the inset taken in section #~5000 (shown in *Figure 2—figure supplements 3b4*, magnified in *Figure 2—figure supplements 3d4*).

The experimenter observed the sectioning between sections #4000 and #5000 and apart from the few sections described below around the restart, no sectioning miss was observed: for all sections, both the magnetic and tissue portions were fully cut and remained connected to each other. No section tear was noticed.

For the restart, sectioning was interrupted at section #~4000 and the last cut section was collected. After a sham MagC collection and a 20 hr break, the first cuts after the restart were manually collected and are shown in *Figure 2—figure supplement 3c*. An horizontal band of same dimensions as described above was acquired in sections #~4000 and in the first seven restart cuts. Careful manual inspection of the EM imagery hinted at no material loss across the restart, which was confirmed by the successful manual tracing of a few small neuronal processes across the restart (shown in *Figure 2—figure supplement 3b,e,f,g*).

## Discussion

### MagC as a new CLEM platform

Since the introduction of Array Tomography in 2007 by the Smith laboratory (*Micheva and Smith, 2007*), volumetric postembedding correlative light and electron microscopy has been performed on manually collected ultrathin sections on flat substrates. Here, I magnetically collected hundreds of consecutive sections of nominal thickness 50 nm directly onto silicon wafers, offering both excellent fluorescent LM (no autofluorescence, easy to immunostain and coverslip) and excellent EM (no charging thanks to good conductivity, very flat substrate).

The high packing density of the sections collected with MagC has allowed the imaging of more than 500 sections from a single piece of silicon wafer, as small as about 2 cm x 2 cm. Such substrates are easy to handle, process, load into microscopes, and store. The high packing density presented advantages for postembedding immunostaining: a few drops of liquids were easily and conveniently deposited at the wafer surface to simultaneously stain hundreds of sections. The packing density also enabled seamless automated light and electron microscopy imaging: after loading the small silicon wafer chip into the microscope of interest and after a simple wafer coordinate calibration, automated acquisition was performed for several hours to several days without interruption.

### Block orientation and use of a dummy

I tested different block orientations (magnetic/tissue parts in left/right, bottom/top, top/bottom and oblique orientations) and found that cutting quality was best when the tissue was placed at the bottom and the magnetic resin at the top. The learned precept is that the most important part of the block, the tissue part, should be cut first. However, in this configuration and with nominal cutting thicknesses below about 60 nm, I noticed that sections tended to go back slightly towards the knife edge at the end of the cutting of some sections, so that the top of the cut section covered the knife edge. This covering tended to impair the sectioning quality (compression) of the bottom of the next section, which was the precious tissue part. To solve this issue, I glued a dummy piece of heavy metal-stained resin-embedded brain tissue at the top of the block, which prevented the knife-covering effect.

### Quality of the section cutting

A potential concern with MagC is that the iron oxide nanoparticles might damage the fragile diamond edge, thus shortening its longevity compared to that when sectioning a block without magnetic resin. Three tears were found in Dataset 1, spanning nine sections in total, and a single tear was found in the magnetic portion of one section in Dataset 2. In the additional cutting experiment with 5000 sections, no tear was noticed among the ~1000 sections observed between sections #4000 and #5000.

Overall, neither in Datasets 1 and 2 nor in the 5000 sections experiment did major section quality issues arise from knife defects potentially resulting from the magnetic nanoparticles. The maximum number of magnetic sections that can be cut reliably on a same knife slot at a given thickness

remains to be assessed, but the experiments showed that MagC is well suited at least for moderate (500 sections) and large-scale experiments (5000 sections).

## Quality of the section collection

The folding or warping of sections during deposition onto a substrate is a common ultramicrotomy concern. These unwanted effects typically arise when the water surface shows strong convexity profiles produced by hydrophobic substrate portions and walls, as sketched in *Figure 2—figure supplement 5f*. The three main precepts to avoid folding or warping drawn from my own experience and from the literature (*Kubota et al., 2018*; *Horstmann et al., 2012*; *Burel et al., 2018*; *Koike et al., 2017*; *Templier and Hahnloser, 2019*; *Harris et al., 2006*; *Wacker et al., 2016*; *Lee et al., 2018*) are to handle sections far from walls, to have a low contact angle between the water surface and the substrate, and to heat the substrate. These three precepts were followed in MagC by maintaining the sections in the center of the knife boat, by making the silicon wafer hydrophilic with a plasma treatment, and by heating the wafer with heating pads placed below the knife boat. As a result, the absence of fold in the collected sections was confirmed in the LM wafer overviews (*Figure 1c*, *Figure 1—figure supplement 1*) and when browsing the correlative LM-EM stacks.

## Automation of MagC and comparison with ATUM

How far can MagC be automated? Magnetic augmentation is unlikely to become an automated procedure even though it is not a complicated manual one. It requires manual handling of the tissue and magnetic blocks, and of the easy-to-use mounting helper device (*Figure 1—figure supplement 3*). Final trimming of the block requires solely moderate ultramicrotomy skills. In comparison, ATUM solely requires standard ultramicrotomy trimming.

Thanks to the ionizer, which gently moves floating sections away from the diamond, sectioning for MagC was fully automated without user intervention. The short sectioning durations for Datasets 1 and 2 (up to only about 1 hr) did not require compensation for water evaporation, but this can be automated too (*Kasthuri et al., 2015*).

Compared to ATUM, the decoupling of sectioning from collection in MagC on one hand removes the need for a failproof collection device running concurrently with sectioning, but on the other hand, maintains the risk of a sudden catastrophic failure until collection is finished (e.g. if the magnet were to touch the water's surface, hundreds of sections could potentially be lost simultaneously). Instead, sections that are collected continuously with ATUM can be gradually considered safe almost immediately upon sectioning.

In the two main datasets presented here, detaching a few sections from the bath walls after sectioning was performed manually with an eyelash probe, which presented a small risk of damaging sections and required moderate ultramicrotomy skills. Blowing air manually with a Pasteur pipette seems to be similarly effective and inherently safer, and it could be automated.

The robotic actuation of the moving magnet was manually controlled and could be automated with a predefined scan path. Similarly, the motorized control of the water level and the magnet height was manual and could also be automated. Overall, if section detachment of a few sections from the walls could be automated, then the collection procedure starting from cutting onset until the sections are positioned on the wafer would be fully automated.

ATUM, unlike MagC, requires the additional manual cutting of the tape into strips and its gluing onto wafers. ATUM and MagC both require the acquisition of overview imagery of the sections on substrates for localization. Automated section localization is, at present, still aided by manual input (*Hildebrand et al., 2017*) but could be soon fully automated with sufficient precision, given that similar tasks of finding numerous instances of objects in microscopy images are convincingly reaching full automation (*Hollandi et al., 2019*). Therefore a single fully automated post-collection procedure can be foreseen for MagC, consisting of wafer overview acquisition, section localization, bead imagery acquisition, and section order retrieval upon insertion of a wafer into a programmatically controlled light microscope.

MagC might therefore be fully automated with the exception of three manual steps: manual magnetic augmentation, manual initiation of sectioning, and final insertion of a wafer into a light microscope for automated section mapping and ordering.

## Upscaling

Can MagC reach the axial depths and volumes currently obtained with other volumetric EM techniques (*Kornfeld and Denk, 2018*), such as the ~15,000 ATUM-collected sections reported by *Hildebrand et al. (2017)*?

The first route to collecting significantly more than the ~500 sections of Dataset 1 is to collect more sections simultaneously covering a larger area at the water surface. For this, larger baths such as that in *Video 1* or even larger could handle many thousands of sections. Concerning the actuation, a conservative claim drawn from the experience of developing MagC is that small magnets (up to about ~20 mm in diameter) can carry sections that occupy an area that is roughly the same size as the area of the magnet (50 nm thick sections,~50/50 ratio of tissue to magnetic resin, 8% magnetic particle concentration). Sections did not accumulate in the center of the magnets, but rather at their periphery, as seen in *Figure 1c* and *Figure 1—figure supplement 1*. This border effect, characterized by leaving a central portion free of sections, was also observed with larger magnets, indicating a scaling of the collection area with the magnet perimeter instead of, perhaps counter-intuitively, its area. Tuning the magnetic field of permanent magnets with special arrangements such as Halbach arrays or using mobile magnets are options to increase the collection area.

A second route for upscaling consists of collecting more sections on several wafers, either simultaneously or sequentially.

For simultaneous collection onto more than one wafer, very large baths could accommodate multiple large wafers and multiple robotically actuated magnets could parallelize the collection procedure with a final water withdrawal, resulting in the simultaneous collection onto several wafers.

For the sequential collection of sections onto more than one wafer, MagC procedures (sectioning, water removal, substrate heating and removal) would be performed one after the other. The success of this approach depends on a core issue, which in part motivated the invention of MagC: long sectioning interruptions impair sectioning quality upon restart. This half a century old restart issue remains unaddressed in the literature and goes beyond the scope of the presentation of the MagC invention here. Only recently have two laboratories, to my knowledge, built precise closed-loop knife positioning systems that could be used to investigate the general restart issue (Briggman [*Briggman, 2015*] and Hesse [iTome, https://vimeo.com/album/4102790]) labs). Even without this equipment, an additional experiment was performed with a commercial ultramicrotome to assess the behavior of a magnetic block after a sectioning restart. The continuity of the collection was demonstrated by manual inspection and by the seamless tracing of a few small neurites across the restart. Nevertheless, this continuity required the manual collection of four sections (#1,#2, #3, #6) that lacked or were disconnected from the magnetic portion upon sectioning, while it would have been possible to collect the other sections with the standard MagC procedure. This experiment did not characterize in depth nor solve the longstanding ultramicrotomy restart issue but at least demonstrated that handling a magnetically augmented block does not seem to deviate much from handling a standard block as regards sectioning restart. If this single experiment is representative, then the optimal implementation of a sequential MagC procedure would be as follows: after magnetic accumulation of almost all sections to a central place, the experimenter would have to collect manually the non-magnetic tissue portions floating around at the surface (in the restart experiment shown here, it would have been the four cuts #1, #2, #3 and #6), before performing the standard MagC procedure. This implementation would require both good ultramicrotomy skills so that these manual collection operations are performed reliably and some adjustments to the image assembly pipeline to deal with partial sections. Also, solving the general restart issue for regular blocks might come close to solving this issue for magnetically augmented blocks.

A third route for the upscaling of MagC is to collect thicker sections and to submit them to cycles of broad (*Templier, 2018*) or gas cluster (*Hayworth et al., 2019*) ion beam milling and EM imaging.

Finally a fourth route, changing the magnetic/tissue ratios, might only give marginal improvements. On one hand, reducing the magnetic area of augmented sections would lead to a final section density increase (though less than two-fold given the current ~50/50 tissue/magnetic ratio), but issues arising from weaker magnetic forces or from the use of higher magnetic particle concentrations (with potentially aggregates damaging the diamond edge) would have to be dealt with. On the other hand, increasing the magnetic area of augmented sections might marginally

increase the attraction range of a magnet and thus the collection area, but it would probably barely compensate, if at all, for the decrease of the tissue/magnetic ratio.

Regarding the sample block sizes, the edge lengths of the magnetically collected sections reported here are in the range 0.5–1.4 mm (Dataset 1: 0.5 × 0.5 mm; Dataset 2: 0.8 mm x 1.4 mm; MultiSEM: 1.1 mm x 1.2 mm; Video experiment: 0.9 mm x 1.1 mm). For blocks with much larger cross-sectional areas, such as whole small mammalian brains (*Mikula and Denk, 2015*), the magnetic actuation should in theory be easier because the actuation force scales with the magnetic area, while the friction force scales with the section side length (*Palagi et al., 2011*), but the ability to section such large blocks has not yet been tested.

Uneven section thickness in serial sectioning is common when no enclosure around the ultramicrotome is used (*Harris et al., 2006*) and occasional missed cuts were observed with a nominal cutting thickness of 30 nm. Most of the work on MagC was performed with a nominal thickness of 50 nm and a definitive lower limit for section thickness was not thoroughly assessed.

## Conclusion

In conclusion, MagC solves the challenge of collecting hundreds of serial ultrathin sections with a high packing density directly onto silicon wafers. I expect MagC to be used in high-throughput volumetric microscopy beyond connectomics for ultrastructural biology in general. Combined with broad (*Templier, 2018*) or gas cluster (*Hayworth et al., 2019*) ion beam milling and next-generation multibeam EM, MagC could become an ideal platform for large-volume EM connectomics.

# Materials and methods

## Animal experiments

Animal experiments were approved by the Veterinary office of Canton Zurich (207/2013). Two zebra finches were anesthetized with isoflurane and placed in a stereotaxic device. Fluorescent tracers were bilaterally injected (0.5–1 µL) into different areas (*Oberti et al., 2010*) as described in *Tables 1*, *2* and *3*. Three to five days after tracer injection, the animals were sacrificed by perfusion fixation with fixative concentrations of 2% formaldehyde and 2.5% glutaraldehyde in buffer with 0.1M cacodylate and 2 mM calcium chloride (referred to as cacodylate buffer). The brain was extracted and slices of 150 µm thickness were cut with a vibratome (Thermoscientific, #Microm HM650V) in cold cacodylate buffer.

Portions of the slices containing the nucleus HVC were dissected out with a surgical scalpel and processed similarly as in the protocols described by *Deerinck et al. (2010)* and *Tapia et al. (2012)*. The sections were washed with cacodylate buffer, stained with heavy metals (2% osmium tetroxide reduced with 1.5% potassium ferrocyanide, washed in ddH2O, 1% thiocarbohydrazide, washed in ddH2O, 2% osmium tetroxide, washed in ddH2O, 1% uranyl acetate at 4 °C overnight, washed in ddH2O, 0.6% lead aspartate, washed in ddH2O), dehydrated with increasing ethanol concentrations (50%, 70%, 80%, 90%, 95%, 100%, 100%), infiltrated in epoxy durcupan resin (10 g component A/M, 10 g B, 0.3 g C, 0.2 g D), and finally cured in an oven at 52°C for 48 hr.

**Table 1.** Coordinates of adult male zebra finch nuclei targeted with tracer injections.

|  | RA | AreaX | Avalanche |
| --- | --- | --- | --- |
| Head angle (degrees) | 65 | 45 | 45 |
| Pipette angle (degrees) | 45 | -20 | 0 |
| Anterior-Posterior (mm) | 3 | 6.45* | 1.8 |
| Media-Lateral (mm) | 2.45 | 1.55 | 2 |
| Dorso-Ventral (mm) | 1.3 | 2.95 | 1.05 |

*with a 0 degree pipette angle

DOI: https://doi.org/10.7554/eLife.45696.018

**Table 2.** Characteristics of the two presented datasets.
BDA: biotinylated dextran amines

| Dataset | Section number | Anatomical region | Tracer | Injection site | Primary antibody | Secondary antibody | EM size (µm x µm) | EM dwell time (ns) | Pixel size (nm) |
|---|---|---|---|---|---|---|---|---|---|
| 1 | 507 | HVC | Alexa 488 FITC Texas red | RA AreaX Avalanche | Rat anti-488 Mouse anti-FITC Goat anti-rhodamine | 488 anti-rat 647 anti-mouse 546 anti-rhodamine | 275 x 205 | 820 | 8 |
| 2 | 203 | Dorsal RA | BDA | Caudal RA | mouse anti-BDA | 647 anti-mouse | 185 x 140 | 6000 | 8 |

DOI: https://doi.org/10.7554/eLife.45696.019

## Resin preparation

To produce a magnetically augmented sample block with homogeneous cutting properties, a magnetic resin was sought that ideally had the same resin formulation as that used for tissue embedding (durcupan), or at least similarly based on epoxy, as well as well-dispersed superparamagnetic nanoparticles. A literature search with these criteria yielded only the publication by *Puig et al. (2012)*, which I decided to reproduce without modifying the epoxy formulation, using the highest reported particle concentration (8% wt) with well-dispersed particles. In addition, a fluorescent dye and fluorescent beads were added. The procedure is as follows.

800 µL of superparamagnetic iron oxide nanoparticles (CAN Hamburg, #SMB-0–038, 10 mg/mL) dispersed in tetrahydrofuran (Sigma-Aldrich, #401757) was mixed for a few seconds with 28 µL oleic acid (Sigma-Aldrich, #O1008) in a small glass vial (VWR, 66030–668) using a glass stirring rod and was left overnight in a fume hood. Then 0.1 g of the epoxy resin DGEBA (Diglycidylether of Bisphenol A, #D3415 Sigma Aldrich) and 5 µL BDMA (Sigma-Aldrich, #185582) were added to the vial. In addition to the original protocol published by *Puig et al. (2012)*, fluorescent particles (Cospheric, mean diameter 2 µm, #FMG, #FMR, 0.2% and 1% wt concentration in Datasets 1 and 2, respectively) and coumarin dye (SigmaAldrich, #257370, 7-Amino-4-methylcoumarin, 0.5% wt concentration) were added to the resin mixture. The vial was then placed on a 90 °C hot plate for 15–20 min while its content was manually stirred with a glass stirring rod until the mix became homogeneous. The resin mixture was poured between a glass slide (bottom) and a piece of aclar sheet (top), both coated with mold separating agent (#62407445, Glorex, as used by *Knott et al., 2011*). A PDMS spacer of about 600 µm thickness surrounded the resin and a small weight was put on top of the aclar sheet for flattening. The resin was cured for 6 hr at 70 °C.

## Block augmentation

For block augmentation, a piece of magnetic resin and a dummy were successively glued to the sample of interest using the durcupan formulation described above for brain tissue preparation. The execution details of the procedure are described in *Figure 1—figure supplement 3*.

## Section collection

The collection procedure is described in the main text. The custom diamond knife with a bath of dimensions 55 mm x 44 mm (35 degrees clearance angle, now commercially available, #Ultra ATS, Diatome, Switzerland) was placed in an ultramicrotome (Leica, UC6) with a 0 degrees knife angle in the ultramicrotome holder. Sections were produced at the water surface at the rate of about eight

**Table 3.** Tracer-antibody library.
LT: Life Technologies. VL: Vector Laboratories. JI: Jackson Immunoresearch.

| Antigen | Alexa 488 LT #D-22910 | FITC LT #D-1820 | Texas Red LT #D-3328 | BDA LT #D-1956 |
|---|---|---|---|---|
| Antibody species | rabbit (LT #A-11094) rat (Biotem #custom) | mouse rabbit (LT #A-889) | goat (VL #SP-0602) | mouse (JI #200-002-211) |

DOI: https://doi.org/10.7554/eLife.45696.020

sections/min using a vertical sectioning speed of 0.4 mm/s in the cutting window, and using the 'fast' return setting of the ultramicrotome arm outside the cutting window.

The ionizer (Leica, #EM Crion), placed a few centimeters away from the knife edge, tended to create a very soft air current that gently brought sections away from the knife edge. This fortuitous feature prevented clogging of sections at the knife edge that could have impaired sections.

The water level in the bath was set with a motorized syringe pump (KDScientific, #210). The setup, shown in *Figure 1b*, consisted of a three-axis motorized actuator (Thorlabs, #LTS150/M, #PT1/M-Z8) carrying an aluminum plate with a goniometer (Thorlabs, #GN2/M) screwed down at its extremity, facing down. A cylindric Neodymium magnet (Supermagnete, cylindrical, 15 mm diameter, 8 mm height) was magnetically anchored to a steel plate screwed to the goniometer. The orientation of the magnet was adjusted with the goniometer in order to align its bottom surface parallel to the water level.

Silicon wafers (Ted Pella, #16015) were cleaved to approximately 40 mm x 45 mm chips, hydrophilized with oxygen plasma (1 min, 25 mA, Emitech #K100X) and placed in the knife bath at an ~ 2 degree angle relative to the water level thanks to asymmetrically stacked microscopy coverslips below the wafer chip, as sketched in *Figure 2—figure supplement 5a*.

## Postembedding staining

### Immunostaining

I deposited and exchanged staining solutions manually with graduated pipettes on the sections collected on flat substrate. All steps were performed at room temperature. The blocking solution was:

1% Baurion BSA-c, 0.05% Tween (*Collman et al., 2015*) in TBS pH 7.4. The detailed procedure was:

1. Blocking — blocking solution — 2 × 10 min
2. Primary antibody incubation — 1:50 in blocking solution — 1.5 hr
3. Washing — TBS — 4 × 5 min
4. Secondary antibody — 1:100 in blocking solution — 1 hr
5. Washing — TBS — 2 × 5 min
6. Washing — dH$_2$O — 2 × 5 min
7. Drying with hand dust blower (Bergeon #30540)
8. Air drying — 5 min

Proceed to fluorescent imaging within the next few hours to avoid decay of staining as reported by *Micheva et al. (2010)*, Fig. Sup. S3.

### Heavy metal post-staining

Heavy metal post-staining was performed by exposing sections on wafer to a few drops of 2% aqueous uranyl acetate, then to a few drops of Reynold's lead citrate (lead 4.4% wt concentration), both for 90 s. Between the two stains and after the second stain, the entire piece of wafer was immersed consecutively in three small Petri dishes of double distilled water for 30 s each. After the second washing, the wafer was dried with a manual air blower.

## Section segmentation

The sections on wafer acquired with low-resolution LM (5x air objective, *Figure 1c*, *Figure 1—figure supplement 1*) were segmented semi-automatically with help of the Trainable Weka Segmentation plugin (*Arganda-Carreras, 2016*) in Fiji/TrakEM2 and custom scripts.

## Section order retrieval with fluorescent beads

### Metric

A metric was defined to assess the quality of the reordering process based on the imagery of fluorescent beads. This metric requires the knowledge of the ground truth order, which I obtained from the section order retrieval performed with EM imagery, and which I call the 'EM order'.

For each section of a reordered dataset, a cost is given to the link between the given section and the next one. The cost is equal to the difference of the indices of the sections in the ground truth order, minus one. For example, the links of the order 1-2-3-4-5-6-7-8 have the costs 0,0,0,0,0,0,0, so do the links of the 8-7-6-5-4-3-2-1 order, while the links of the order 1-2-4-5-3-8-6-7 have the costs

0,1,0,1,4,1,0. A single flip such as 1-**2-4**-3-5-6 has the cost 0,1,0,1,0,0. The frequency of these costs gives an estimate of how precise the reordering is.

## Section order retrieval

After preprocessing, the fluorescent bead imagery ('Normalize local contrast' Fiji plugin, thresholding), the center location of the beads was extracted (Maxima Finder) for each fluorescent channel. The locations of the beads from the two fluorescent channels were merged into a single final channel. I computed a dissimilarity value for every pair of sections. For each pair of bead center sets, descriptor matching was performed (using the descriptor-based bead alignment available in Fiji [*Preibisch et al., 2009a*]). If no geometric match was found for a given pair of sections, then the dissimilarity value was set to a fixed large number. If a geometric match was found, then a matching affine transform was computed and applied to the first bead set, thus bringing the pair of bead sets into a same coordinate system. In this common coordinate system, the bead centers contained outside a central bounding box were excluded from further calculations to avoid considering beads that are present in one section but not in the other one due to a limited field of view and due to the different orientations of the section. The pair of remaining bead sets was then matched again with the descriptor-based tool. For each match, that is each pair of two matching beads, the absolute difference of the diameters of the matching beads was computed. The dissimilarity of two sections was then defined as the sum of these diameter differences across all matching beads. A traveling salesman problem was formulated using the dissimilarities as distances between nodes of a graph, and the problem was solved with the Concorde solver (*Applegate et al., 2003*).

## Section-order retrieval with EM

An EM section was made of a mosaic of EM tiles ($3 \times 3$ or $2 \times 2$). For a given pair of EM sections, a dissimilarity was computed for each pair of corresponding mosaic tiles and averaged across the tiles to yield the complete dissimilarity between two EM sections. The dissimilarity of two tiles was calculated as follows: an affine transform matching was sought between the pair of images using the SIFT matching algorithms implemented in Fiji. If no affine transform was found, then the pair of tiles was given an arbitrary high dissimilarity. If a transform was found, then it was used to align the two tiles and a normalized cross-correlation was computed in a central box of $2000 \times 2000$ pixels. The value ($2 -$ correlation) was used as the dissimilarity value between the two tiles. When averaging the dissimilarities across tiles for a given pair of EM sections, the non-matching tiles were excluded if other tiles were matching. It made the dissimilarity value more robust to artifacts that may have prevented a match from being found in one of the tiles. As with the beads, an open traveling salesman problem was solved with the computed dissimilarities that yielded the original order, as confirmed with manual inspection of the EM stack.

## Imaging

### Wafer overview

The sections on wafers were imaged in mosaics at low resolution (5x air objective) with widefield reflection brightfield and fluorescent light microscopy (so-called DAPI, GFP, and RFP channels) using a Zeiss Z1 microscope. Tiles were stitched with the Fiji stitching plugin (*Preibisch et al., 2009b*) on the brightfield channel.

### Fluorescent beads

The silicon wafer chip was installed in the holder of an inverted LM microscope (Nikon Ti Eclipse; inverted; Objective — 20x-0.7 NA; illumination — Lumencor, #Spectra). Using a python script controlling the microscope through Micromanager (*Stuurman et al., 2010*; *Edelstein et al., 2014*), landmarks were mapped to generate a rigid transform between the coordinates of the low-resolution wafer overview imagery and the coordinates of the wafer in the microscope stage. The mapping consisted of manually driving the x-y stage to the first two landmarks and clicking a button. The custom software then conveniently placed the stage at the calculated locations of the remaining landmarks for the user to adjust the exact location. After manual calibration of the hardware autofocus (Nikon, #PerfectFocusSystem), fluorescent LM acquisition was performed automatically with the same script controlling the microscope through Micromanager. Depending on the cross-sectional

area of the magnetic resin, a single field of view or a mosaic of fields of views can be acquired at each section. A single field of view was used for the datasets presented here. Hardware autofocus was performed at each field of view. The channels that were used were the standard so-called GFP and RFP for the fluorescent beads, and DAPI for visualization of the coumarin dye.

## Fluorescently stained tissue

Drops of mounting medium (Molecular Probes, #S36937) were deposited on the freshly stained sections and were subsequently covered with standard microscope coverslips of appropriate size to cover the area of collected sections. The coverslip was maintained in place with a pair of small magnets (one below the wafer, one above the coverslip, supermagnete, #S-04–01 N, neodymium cylinder, 1 mm thickness, 4 mm diameter).

The silicon wafer substrate was placed in the holder of an inverted fluorescent light microscope (Nikon Ti Eclipse, illumination: Lumencor, #Spectra). Landmarks were mapped with my custom software in two steps: first with a 20x dry objective, then with a high-magnification 63x oil-immersion objective. The reason for this two-step procedure is that the use of an inverted microscope prevented intuitive manual navigation to find the initial first two landmarks, which would have been a cumbersome task using solely a high-magnification objective. After successful mapping with the 20x air objective, immersion liquid was added onto the coverslip and the holder was put back in the same position. The script then guided the user through all landmarks, so that they only had to adjust the high-magnification objective precisely at the landmark locations suggested by the low-magnification objective mapping. As with the bead imaging earlier, an affine transform was computed to transform the coordinates from the wafer overview imagery to the microscope stage coordinates.

We defined all imaging parameters in our custom python software: fluorescent channels (GFP, RFP, YFP, brightfield), exposure times (500 ms and 1 ms for {GFP, RFP, YFP} and brightfield, respectively), z-offset per channel (relative to the autofocus channel reference), imaging grid ($2 \times 2$ and $1 \times 1$ mosaic for Datasets 1 and 2, respectively). Multichannel mosaics were automatically imaged for all sections with hardware autofocus activated at each field of view.

After LM imaging, the pair of small magnets was gently removed before removing the cover slip while taking care that the immersion oil did not come into contact with the wafer. The wafer was immersed three times for 5 min each in a small dish of double-distilled water to wash away the mounting medium. The wafer was finally dried at room temperature with a hand blower.

## Electron microscopy

The wafers were mounted on standard EM stubs (Tedpella, #16111, #16144) with double-sided carbon tape (Tedpella, #16084–1, #16084–2) and inserted into a scanning electron microscope (Zeiss, Merlin).

The sample was cleaned for 10 min with an in-chamber air plasma (Evactron, Zephyr model 25 plasma cleaner) to minimize carbon contamination during imaging with the secondary electron inlens detector.

As with LM imaging, four previously defined landmarks were mapped onto the wafer with the three-axis stage of the EM to provide the x-y coordinates (z-axis remained fixed), angle and working distance of each section. The scanning angle of the beam was rotated according to the section angle so that each section was acquired with the same orientation.

Mosaics of $3 \times 3$ (Dataset 1) and $2 \times 2$ (Dataset 2) tiles were acquired for each section. An autofocus-autostigmation-autofocus sequence was performed at the center of each mosaic. To avoid performing that sequence on a low-contrast area, such as a large cell body or a blood vessel, a subregion was automatically selected around the center of the mosaic that contained maximal contrast (as determined by the intensity after applying an edge filter). The main EM imaging parameters were: two keV incident energy, secondary electron inlens detector, 800 pA current probe, 3.5 mm working distance, and 750 ns and 6000 ns dwell time for Datasets 1 and 2, respectively.

## Multibeam EM

In another small experiment, 15 sections were collected with MagC on a small silicon wafer chip (*Figure 1—figure supplement 5a*). One section was acquired at 4 nm/pixel resolution with a 91-beam

Zeiss MultiSEM scanning electron microscope and 400 ns dwell time (there was no intermediate immunostaining and no heavy-metal poststaining).

In another experiment, after the single-beam EM acquisition of Dataset 1 (that is, after the entire CLEM pipeline including immunostaining, LM, washing, poststaining, single beam EM), the wafer chip (*Figure 1c*) was submitted to wafer-wide homogeneous broad ion beam milling (Veeco Nexus IBE350, two degrees glancing angle, 12 s, 0.8 kV ion energy, 10 rpm rotation) resulting in homogeneous section etching of about 25 nm (Templier, in preparation). One section was then acquired with a Zeiss MultiSEM, with the same imaging conditions as mentioned above.

## Data assembly

The brightfield channel of the LM imagery was used for the stitching, alignment and registration operations (with an initial 'Normalize local contrast' from Fiji with blocks of about 100 pixels x 100 pixels) performed in Fiji. The stitching was then propagated to all fluorescent channels. Stitching and alignment (*Saalfeld et al., 2012*) of the EM imagery was done with custom scripts in TrakEM2.

For cross-modality registration, the stitched mosaics of the brightfield channel were preprocessed with local contrast enhancement and Gaussian blurring. The EM counterpart mosaics were downsampled to exhibit roughly the same pixel size as the LM imagery and further preprocessed with local contrast enhancement. The LM brightfield and EM imageries then exhibited a similar appearance so that corresponding SIFT features (*Lowe, 2004*) could be computed across the two modalities (*Figure 2—figure supplement 2*). Moving least squares transforms (*Schaefer et al., 2006*) were computed on the basis of these matching SIFT features using Fiji. The transforms were then upsampled and applied to all fluorescent channels of the LM imagery in the TrakEM2 plugin to yield a volumetric correlative LM-EM stack.

For visualization purposes, the correlative LM-EM imagery was converted to the neuroglancer format and hosted online for convenient in-browser visualization and annotation (see Methods, paragraph K). Datasets 1 and 2 are available at https://neurodata.io/data/templier2019/.

## Conversion of correlative LM-EM imagery for neuroglancer

The EM imagery assembled in TrakEM2, along with all transforms (affine, elastic and moving least squares), was converted into a Render (https://github.com/saalfeldlab/render) project (*Zheng et al., 2018*) with custom scripts and the TrakEM2 converter script of the Render project. Similarly, TrakEM2 projects were created for each LM channel that contained stitching and moving least square transforms. These TrakEM2 projects were converted to separate Render projects. The imagery of the EM and LM Render projects was rendered to files using a custom script and the Render script for mipmap creation (render_catmaid_boxes). With a custom script, these mipmaps were then used to create chunks at different resolutions in the 'precomputed format' of Neuroglancer (https://github.com/google/neuroglancer). The chunks were uploaded to an online cloud storage service (Google storage) and an instance of the Neuroglancer software hosted online (neurodataviz from the MICrONS project) was used to visualize the data. The EM imagery and each fluorescent LM channel were added into a neuroglancer session as separate data sources. After online visualization with neuroglancer, stacks of correlative imagery were fetched using the cloud-volume library (https://github.com/seung-lab/cloud-volume). Neurite tracings were performed in neuroglancer (line annotations).

## Sectioning quality and restart experiment

The following experiment was performed to assess the potential knife damage caused by magnetic particles and whether next-day sectioning restart of magnetically augmented blocks leads to section losses.

A sample was positioned on a new knife portion that had not been used since resharpening by the diamond manufacturer (35 degrees clearance angle, Diatome, Switzerland). 2000 cutting cycles were performed (50 nm thick, about 1 mm/s cutting speed) and five sections were manually collected corresponding approximately to section numbers #100, #500, #1000, #1500, #2000. Sectioning was interrupted, the floating sections were discarded to free up space at the water surface (by sweeping a wooden stick over the water surface), and the knife edge was cleaned with styrofoam according to the manufacturer recommendations. Sectioning was resumed for 2000 additional

cutting cycles during which four sections were manually collected (#2500, #3000, #3500, #4000). The last cut section was collected and is referred to as section number #~4000. The floating sections were again discarded and a sham MagC collection was performed, consisting of removing the water and activating the heating pads for 1 hr. The knife remained locked in place in the ultramicrotome.

About 20 hr later, the sample block was moved back a few microns away from the knife to prevent an unwanted touch at restart. The ultramicrotome was started and the first seven cuts were manually collected, deposited onto a silicon wafer, and inspected with scanning EM. The cutting was interrupted after 15 cuts, while the experimenter kept track of the locations in the bath of the first sections.

Sectioning was resumed for 1000 more sections, during which the experimenter visually inspected sectioning through the binocular of the ultramicrotome and counted failed sectioning attempts. Finally, section number #~5000 was manually collected.

## Video-recorded magnetic collection

A different setup was used for *Video 1* but it had no important differences compared to the setup used for Datasets 1 and 2. The ultramicrotome used was a Reichert-Jung Ultracut E, the custom knife bath was larger (11 cm x 11 cm), a cylindric magnet (15 mm diameter x 8 mm) and a silicon wafer of 100 mm diameter (P-type; Boron-doped; <100> OFF; resistivity 0.1–100 Ohm. cm) were used. Sections were manually blown away from the bath walls with a glass pipette, which was used to produce very gentle puffs of air. 100 consecutive sections were cut with a nominal thickness of 50 nm. The first section had only a small partial magnetic part, which nevertheless remained stuck to the second section, therefore it was still successfully magnetically collected with all of the other sections.

The timeline of the events that occurred during the collection was as follows: 1. Ultramicrotome start (00:02); 2. Cutting (from 00:19); 3. Cutting stopped (06:21); 4. Removal of ionizer (06:32); 5. Magnet scanning (from 07:02); 6. Blowing away two sections from wall (17:25); 7. Blowing away one section from wall (20:42); 8. Water removal (from 27:15); 9. Heating (from 31:33); 10. Wafer pickup (45:41).

## Acknowledgements

I thank N Broguiere and H Gnaeggi for comments on the manuscript and help in the design of the custom knife boat, respectively. I thank members of the Zeiss multiSEM team (Eberle, Nickell, Garbowski) for the multibeam scanning EM experiments. I acknowledge the support of the Scientific Center for Optical and Electron Microscopy ScopeM of the Swiss Federal Institute of Technology ETHZ, and of the Center of Micronanotechnology, École polytechnique fédérale de Lausanne EPFL. I acknowledge support from a Bridge Proof of Concept fellowship from Innosuisse-Swiss National Foundation. I thank the NeuroData team for hosting the datasets.

## Additional information

### Competing interests
Thomas Templier: A patent application has been filed by ETH Zurich (EP3171150A1).

### Funding

| Funder | Grant reference number | Author |
| --- | --- | --- |
| ETH Zürich | Foundation ETH Grant 42 15-1 | Thomas Templier |
| Innosuisse | Bridge Proof of Concept 173825 | Thomas Templier |
| Schweizerischer Nationalfonds zur Förderung der Wissenschaftlichen Forschung | Bridge Proof of Concept 173825 | Thomas Templier |

The funders had no role in study design, data collection and interpretation, or the decision to submit the work for publication.

## Author contributions
Thomas Templier, Conceptualization, Data curation, Software, Formal analysis, Funding acquisition, Validation, Investigation, Visualization, Methodology, Writing—original draft, Writing—review and editing

## Author ORCIDs
Thomas Templier  https://orcid.org/0000-0002-0523-5947

## Ethics
Animal experimentation: Animal experiments were approved by the Veterinary office of Canton Zurich (207/2013).

## Decision letter and Author response
Decision letter https://doi.org/10.7554/eLife.45696.027
Author response https://doi.org/10.7554/eLife.45696.028

# Additional files

## Supplementary files
• Transparent reporting form
DOI: https://doi.org/10.7554/eLife.45696.021

## Data availability
Datasets 1 and 2 are publicly available for online visualization and download at https://neurodata.io/data/templier2019. Code is at https://github.com/templiert/MagC (copy archived at https://github.com/elifesciences-publications/MagC).

The following datasets were generated:

| Author(s) | Year | Dataset title | Dataset URL | Database and Identifier |
|---|---|---|---|---|
| Thomas Templier | 2019 | templier19_dataset_1 | https://neurodata.io/data/templier2019/ | NeuroData, templier19_dataset_1 |
| Thomas Templier | 2019 | templier19_dataset_2 | https://neurodata.io/data/templier2019/ | NeuroData, templier19_dataset_2 |

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
