## [Decision Letter]

Thank you for submitting your article "MagC, magnetic collection of ultrathin sections for volumetric correlative light and electron microscopy" for consideration by *eLife*. Your article has been reviewed by two peer reviewers, and the evaluation has been overseen by a Reviewing Editor and Eve Marder as the Senior Editor. The reviewers have opted to remain anonymous.

The reviewers have discussed the reviews with one another and the Reviewing Editor has drafted this decision to help you prepare a revised submission.

As you can see below, both reviewers value the innovative approach of the method. However, both reviewers request a more elaborate discussion of the limitations of the approach. At the same time, reviewer 2 lists a series of insufficiencies of the manuscript's presentation, which we expect to be addressed in a revised version.

In particular, the raw data should be available for the reviewers before the manuscript can be accepted. Similarly, the data should be made available (and stay available) for the readers after eventual publication.

Finally, we ask you to resubmit the paper as a Tools and Resources paper, as it is a Methods paper.

*Reviewer #1:*

In the manuscript "MagC, magnetic collection of ultrathin sections for volumetric correlative light and electron microscopy" Templier details a novel solution to a historically vexing problem -how to reliably collect hundreds of ultrathin sections for serial electron microscopic 3D reconstruction.

Traditionally electron microscopists will carefully trim and prepare the sample block so that the resulting ultrathin sections stick together in long trains in the order they were cut while floating in the diamond knife boat (e.g. Harris et al., 2006). These trains of sections can be collected on a silicon wafer by carefully nudging them with an eyelash while water is drained from beneath them (e.g. Burel et al., 2018). Templier's method avoids the complications involved in ensuring trains of sections reliably form while cutting. It also avoids complications involved in ensuring that these trains stay in position above the silicon wafer as the water is drained. He does this by eschewing train formation altogether, instead trimming the block to a point and positioning an air current source to ensure that sections do not stick together to form trains. Crucially, Templier describes how to augment the sample block with superparamagnetic nanoparticles (along with fluorescent beads) prior to ultrathin sectioning so that each individual ultrathin section will be attracted by a permanent magnet which, following completion of sectioning, is positioned 1 mm above the water's surface and swept over the boat to collect and concentrate all sections directly above a submerged silicon wafer. The water is then withdrawn, and heating pads are engaged, causing the ultrathin sections to adhere to the silicon wafer supposedly without wrinkle formation. Templier describes how section ordering can be restored computationally by imaging the fluorescent beads in LM and calculating the correlation matrix of all collected sections. Templier proceeds to demonstrate his method with two correlative LM and EM datasets.

Importance: Dense cellular connectomics is important to neuroscience research and multibeam scanning electron microscopes are being used by an increasing number of labs to push to larger volumes. Since these multibeam SEM perform best when sections are collected directly onto silicon wafers, Templier's MagC method provides a much needed solution for such section collection.

Novelty: Other solutions exist to the problem of collecting ultrathin sections on silicon wafers (e.g. Burel et al., 2018) but Templier's MagC method has the potential to be easier and more repeatable as it avoids many of the manual steps.

Effectiveness: Templier provides two datasets (507 and 203 sections, 50 nm section thickness) as evidence of the effectiveness of the MagC technique. Aside from minor artifacts, these datasets show that neurites can be traced throughout and large immunolabeled processes can be identified in LM and correlated with the EM.

Concerns: A more comprehensive discussion of problems that can occur and steps to avoid them would be helpful.

Minor comment:

A few more details about Figure 1—figure supplement 5 would be helpful. This is supposedly a single multibeam SEM montage from a larger MagC collected run. How many sections were collected in this run? Is this image representative of the overall quality? Was multibeam SEM imaging also compatible with the LM immuolabeling shown in the other datasets?

*Reviewer #2:*

Templier describes a method to automate the collection of serial sections by embedding magnetic nanoparticles in a sample block and collecting cut sections with a moveable permanent magnet. This is a clever idea and addresses some of the issues with the competing ATUM technology. However, since the manuscript is meant to describe a new methodology, there are several limitations with the manuscript in its current form.

1) The description of the preparation of the magnetic resin is missing. Since this is a key component to the method, it should be explicitly described instead of only citing Puig et al. Details such as the instrumentation used, dispersion method, why a different resin was chosen for the iron nanoparticles, and any optimization of particle concentration that would allow a reader to replicate the procedure are conspicuously missing from the manuscript.

2) The format of the article text reads more like a hastily prepared lab report than a scientific manuscript. The supplementary figure legends contain oddly placed methodological details such as subsections “Metric for order retrieval and order retrieval for data set 1”, “Immunostaining protocol” and “Text 1: conversion of correlative LM-EM imagery for neuroglancer” interspersed with supplementary figures. These details should be in the main Materials and methods section I think, or at least less confusingly organized.

3) Critical issues are not addressed such as a) exploring whether block face size effects the ability of the magnet to collect sections, b) whether the magnetic particles degrade the cutting properties of the knife for thousands of sections, c) can sections less than 50nm thick be collected with the magnet at the nanoparticle density described?

4) A benefit of ATUM is that the collection of thousands of sections can be collected without user intervention. The articles does not address how this process works for MagC. Can sectioning be continued after mounting one batch of sections on a wafer without losing a section? What is the error rate associated with restarting the section collection process?

5) Were the sections that needed to be manually detached from the wall of the boat damaged?

6) The logic behind attaching the dummy piece is not sufficiently described. What happens if the dummy piece is not attached?

7) A downloadable image stack is preferable to a youtube video (Video 1) for assessing data quality. I'm not sure what is the scientific content of Video 2, it seems more like a PR video.

8) I have tried to open the location of the datasets with multiple browsers, but have only received ‘site not’ found errors.

9) When trying to visualize the link showing traced neurites I could not see the raw data and received an error.

10) I couldn't find details in the Materials and methods about what type of LM and SEM was used to perform the imaging and under what imaging conditions.

11) Details about how cutting was performed are missing, such as cutting speed, knife angle, clearance angle, etc…

12) A video of the process of collection (demonstrating the snake path of the magnet and dropping the water level) would be very helpful.

13) The main text describes the LM imaging procedure briefly but the Materials and methods do not expand on this. Does the immersion oil on top of the sections simply wash off? Are there any other cleaning steps before heavy metal post-staining?

14) The 'Data analysis' portion of the main text is more focused on data visualization. Some detail is needed regarding the length of traced neurites, etc…

15) Were any sections distorted during the collection process (e.g. folds, tears, warping), particularly during the mounting step down onto the wafer? Was this assayed?

16) Typo subsection “Section collection, last paragraph”, should be 45 mm.

17) The description of 'asymmetrically stacked microscopy coverslips' (subsection “Section collection, last paragraph”) is unclear. A diagram would help to show how the wafer is actually situated in the boat.

18) Figure 1—figure supplement 3 references a panel 'H' that is not in the actual figure.

19) Figure 1—figure supplement 1: The sections are of different intensities and contrast. What is the source of this variability?

20) Figure 2—figure supplement 1 references 6 panels, there are only 4 in the figure.

21) What is the actual contamination in Figure 1—figure supplement 2D. Is this an aggregation of iron nanoparticles?

22) Figure 1—figure supplement 2, there appear to be slivers detached form the sections in panel A. What are these? I don't see the dummy piece in these sections. Is the attachment of the iron in this example the same as presented in Figure 1?

---

## [Author Response]

As you can see below, both reviewers value the innovative approach of the method. However, both reviewers request a more elaborate discussion of the limitations of the approach. At the same time, reviewer 2 lists a series of insufficiencies of the manuscript's presentation, which we expect to be addressed in a revised version.In particular, the raw data should be available for the reviewers before the manuscript can be accepted. Similarly, the data should be made available (and stay available) for the readers after eventual publication.

Please see answers to points 8) And 9) of reviewer #2.

The neurodata team (https://neurodata.io/) told me that they will decide on hosting my datasets online (easy public access with online browsing with neuroglancer) after the acceptance of the paper.

Finally, we ask you to resubmit the paper as a Tools and Resources paper, as it is a Methods paper.

Okay.

Reviewer #1:

[…] Importance: Dense cellular connectomics is important to neuroscience research and multibeam scanning electron microscopes are being used by an increasing number of labs to push to larger volumes. Since these multibeam SEM perform best when sections are collected directly onto silicon wafers, Templier's MagC method provides a much needed solution for such section collection.Novelty: Other solutions exist to the problem of collecting ultrathin sections on silicon wafers (e.g. Burel et al., 2018) but Templier's MagC method has the potential to be easier and more repeatable as it avoids many of the manual steps.Effectiveness: Templier provides two datasets (507 and 203 sections, 50 nm section thickness) as evidence of the effectiveness of the MagC technique. Aside from minor artifacts, these datasets show that neurites can be traced throughout and large immunolabeled processes can be identified in LM and correlated with the EM.Concerns: A more comprehensive discussion of problems that can occur and steps to avoid them would be helpful.

I think that your concern is now addressed with my response to the points mentioned by reviewer #2 and by the Reviewing Editor.

Minor comment:A few more details about Figure 1—figure supplement 5would be helpful. This is supposedly a single multibeam SEM montage from a larger MagC collected run. How many sections were collected in this run? Is this image representative of the overall quality? Was multibeam SEM imaging also compatible with the LM immuolabeling shown in the other datasets?

The following has been added in the Methods and Results, and the figure about multibeam SEM has been extended.

Methods:

“MultiSEM experiments

In another small experiment, 15 sections were collected with MagC on a small silicon wafer chip (Figure 1—figure supplement 5A). One section was acquired at 4 nm/pixel resolution with a 91-beam Zeiss MultiSEM scanning electron microscope, 400 ns dwell time (there was no intermediate immunostaining, no heavy metal poststaining). In another experiment, after the single beam EM acquisition of dataset 1 (that is, after the entire CLEM pipeline including immunostaining, LM, washing, single beam EM), the wafer chip (Figure 1C) was submitted to wafer-wide homogeneous broad ion beam milling (Veeco Nexus IBE350, 2 degrees glancing angle, 12 seconds, 0.8 kV ion energy, 10 rpm rotation) resulting in homogeneous section etching of about 25 nm (Templier, in preparation). One section was then acquired with a Zeiss MultiSEM, with the same imaging conditions as mentioned above.”

Results:

“Multibeam scanning

EM was also used successfully to image magnetically collected sections (Figure 1—figure supplement 5) in two settings: without intermediate treatments after collection (no immunostaining, no mounting, no LM, no poststaining), and after performing the entire CLEM pipeline on the wafer of data set 1 (immunostaining, mounting, LM, washing, single beam EM) and an additional wafer-wide broad ion beam milling of about 25 nm (Templier, in preparation).”

Reviewer #2:

Templier describes a method to automate the collection of serial sections by embedding magnetic nanoparticles in a sample block and collecting cut sections with a moveable permanent magnet. This is a clever idea and addresses some of the issues with the competing ATUM technology. However, since the manuscript is meant to describe a new methodology, there are several limitations with the manuscript in its current form.1) The description of the preparation of the magnetic resin is missing. Since this is a key component to the method, it should be explicitly described instead of only citing Puig et al. Details such as the instrumentation used, dispersion method, why a different resin was chosen for the iron nanoparticles, and any optimization of particle concentration that would allow a reader to replicate the procedure are conspicuously missing from the manuscript.

The paragraph about resin preparation in the Materials and methods section has been extended.

“To have a magnetically augmented sample block with homogeneous cutting properties, a magnetic resin was sought with ideally the same resin formulation as used for tissue embedding (Durcupan) or at least similarly based on epoxy, and with well dispersed small sized superparamagnetic nanoparticles. […] A PDMS spacer of about 600 μm thickness surrounded the resin and a small weight was put on top of the aclar sheet for flattening. The resin was cured for 6 hours at 70C.”

2) The format of the article text reads more like a hastily prepared lab report than a scientific manuscript. The supplementary figure legends contain oddly placed methodological details such as subsections “Metric for order retrieval and order retrieval for data set 1”, “Immunostaining protocol” and “Text 1: conversion of correlative LM-EM imagery for neuroglancer” interspersed with supplementary figures. These details should be in the main Materials and methods section I think, or at least less confusingly organized.

The methodological details mentioned have been moved to the Materials and methods section.

3) Critical issues are not addressed such as:a) Exploring whether block face size effects the ability of the magnet to collect sections.

The following paragraph has been added in the Discussion.

“Regarding the sample block sizes, the edge lengths of the magnetically collected sections reported here are in the range 0.5 mm to 1.4 mm (dataset 1: 0.5 x 0.5 mm; dataset 2: 0.8 mm x 1.4 mm; MultiSEM: 1.1 mm x 1.2 mm; Video experiment: 0.9 mm x 1.1 mm). For blocks with much larger cross-section areas such as whole small mammalian brains (Mikula and Denk, 2015), the magnetic actuation should in theory be easier because the actuation force scales with the magnetic area while the friction force scales with the section side length (Palagi et al., 2011), but the ability to section such large blocks has not yet been tested.”

b) Whether the magnetic particles degrade the cutting properties of the knife for thousands of sections.

Please see answer to reviewer #2 point 4.

c) Can sections less than 50nm thick be collected with the magnet at the nanoparticle density described?

The following paragraph has been added to the Discussion.

“Uneven section thickness in serial sectioning is common when no enclosure around the ultramicrotome is used (Harris et al., 2006) and occasional missed cuts were observed with a nominal cutting thickness of 30 nm. Most of the work on MagC was performed with a nominal thickness of 50 nm and a definitive lower limit for section thickness was not thoroughly assessed.”

4) A benefit of ATUM is that the collection of thousands of sections can be collected without user intervention. The articles does not address how this process works for MagC. Can sectioning be continued after mounting one batch of sections on a wafer without losing a section? What is the error rate associated with restarting the section collection process?

An additional experiment was performed and the following paragraphs (in Methods, Results and Discussion) and a supplementary figure have been added.

Materials and methods:

“Sectioning quality and restart experiment

The following experiment was performed to assess potential knife damage caused by magnetic particles and whether next day sectioning restart of magnetically augmented blocks leads to section losses.[…] Finally, section number #~5000 was manually collected.”

Results:

“In addition, an experiment was performed with a magnetically augmented block (one section shown in Figure 2—figure supplement 3A) with two purposes: 1) assessing knife damage due to the magnetic resin by cutting about 5000 sections with a same knife portion and counting the number of sectioning fails between sections #4000 and #5000, and 2) assessing the continuity of the imagery across a sectioning restart that included a sham MagC collection and a 20 hours long break. The timeline of the sectioning experiment is shown at the right of panel B in Figure 2—figure supplement 3. […] Careful manual inspection of the EM imagery hinted at no material loss across the restart, confirmed by the successful manual tracing of a few small neuronal processes across the restart shown in Figure 2—figure supplement 3B, E, F, G.”

Discussion:

“Automation of MagC and comparison with ATUM

How far can MagC be automated? Magnetic augmentation is unlikely to become an automated procedure even though it is not a complicated manual one. […] On the other hand, increasing the magnetic area of augmented sections might marginally increase the attraction range of a magnet and thus the collection area, but it would probably barely compensate, if at all, the decrease of the tissue/magnetic ratio.”

5) Were the sections that needed to be manually detached from the wall of the boat damaged?

These sections were not damaged. I added the three last words to this sentence “A few sections (about a dozen) that were slightly sticking to the walls of the bath were gently detached with an eyelash and remained undamaged.”

6) The logic behind attaching the dummy piece is not sufficiently described. What happens if the dummy piece is not attached?

The following paragraph has been added in the Discussion:

“Block orientation and use of dummy

I tested different block orientations (magnetic/tissue parts in left/right, bottom/top, top/bottom and oblique orientations) and found that cutting quality was best when the tissue was placed at the bottom and the magnetic resin at the top. […] To solve this issue, I glued a dummy piece of heavy metal-stained resin-embedded brain tissue at the top of the block, which prevented the knife-covering effect.”

7) A downloadable image stack is preferable to a youtube video (Video 1) for assessing data quality.

The stack has been attached to the submission.

I'm not sure what is the scientific content of Video 2, it seems more like a PR video.

*eLife*’s audience is broad and not all readers are familiar with the scales at play with correlative light and electron microscopy. This video gives a summary at a glance of what the new technique provides: many ultrathin sections collected directly onto a small single wafer chip, with magnetic, fluorescent resin and tissue that has been immunostained and imaged with light and finally electron microscopy.

I would be in favor of keeping it but I leave the decision of keeping this video to whoever is in charge to decide.

8) I have tried to open the location of the datasets with multiple browsers, but have only received ‘site not found’ errors.

The url address of the neurodataviz app has been changed during review of the manuscript (from https://viz.boss.neurodata.io to https://viz.neurodata.io, which was not in my control), making the original links obsolete starting from around end of February 2019 (it seems that reviewer #1 still was able to browse the datasets). I apologize for this. The two datasets are now browsable at https://neurodata.io/data/templier2019.

9) When trying to visualize the link showing traced neurites I could not see the raw data and received an error.

It is an unfortunate error that I do not understand, I apologize for not noticing it: the second hyphen character is getting lost when copy-pasting the link from the PDF, despite the fact that the hyphen is visible in the PDF file. This problem is not present with the.docx version of the file. One can either:

- use the link in the article resubmitted in.docx format

- use the link from the original PDF file and manually add the missing hyphen at the beginning of the link in “phd-data”

10) I couldn't find details in the Materials and methods about what type of LM and SEM was used to perform the imaging and under what imaging conditions.

Details about LM and SEM imaging have been added in the Imaging paragraph of the Materials and methods section.

“Imaging

Wafer overview

The sections on wafers were imaged in mosaics at low resolution (5x air objective) with widefield reflection brightfield and fluorescent light microscopy (so-called DAPI, GFP, and RFP channels) using a Zeiss Z1 microscope. […] The main EM imaging parameters were: 2 keV incident energy, secondary electron inlens detector, 800 pA current probe, 3.5 mm working distance, 750 ns and 6000 ns dwell time for data sets 1 and 2, respectively.”

11) Details about how cutting was performed are missing, such as cutting speed, knife angle, clearance angle, etc…

The details have been added in the Materials and methods part to the existing subsection “Section collection”.

“The collection procedure is described in the main text. The custom diamond knife with a bath of dimensions 55 mm x 44 mm (35 degrees clearance angle, now commercially available, #Ultra ATS, Diatome, Switzerland) was placed in an ultramicrotome (Leica, UC6) with a 0 degrees knife angle in the ultramicrotome holder. […] This fortuitous feature prevented clogging of sections at the knife edge that could have impaired sections.”

12) A video of the process of collection (demonstrating the snake path of the magnet and dropping the water level) would be very helpful.

There was no video recording of the collection of data sets 1 and 2. As part of current efforts to scale up the technique, I am now working with a different setup, which I used to produce a video of the process in a small experiment with 100 collected sections. The following Materials and methods, Results, Video 3 and Figure 1—figure supplement 4 have been added.

Materials and methods:

“A different setup was used for Video 3 but it had no important difference compared to the one used for data sets 1 and 2. Ultramicrotome Reichert-Jung Ultracut E, the custom knife bath was larger (11 cm x 11 cm), cylindric magnet (15 mm diameter x 8 mm), silicon wafer 100 mm diameter, P; Boron; <100>-; 0.1-100 Ohm.cm. […] Timeline of the events that occurred during the collection: 1) Ultramicrotome start (00:02); 2) Cutting.… (00:19); 3) Cutting stopped (06:21); 4) Removal of ionizer (06:32); 5) Magnet scanning.… (07:02); 6) Blowing away two sections from wall (17:25); 7)Blowing away one section from wall (20:42); 8) Water removal.… (27:15); 9) Heating.… (31:33); 10) Wafer pickup (45:41).

Results:

“The magnetic collection of these datasets was not video recorded but Video 3 shows another small complete MagC experiment with 100 consecutive 50 nm thick sections, and Figure 1—figure supplement 4 shows the sections on the silicon wafer.”

Video 3: Video-recorded magnetic collection

Video available here: https://youtu.be/o13r-tHT9-c

13) The main text describes the LM imaging procedure briefly but the Materials and methods do not expand on this. Does the immersion oil on top of the sections simply wash off? Are there any other cleaning steps before heavy metal post-staining?

Details of the procedure have been added in the Imaging part of the Materials and methods section:

“Drops of mounting medium (Molecular Probes, #S36937) were deposited on the freshly stained sections and were subsequently coverslipped with standard microscope coverslips of appropriate size to cover the area of collected sections. […] The wafer was immersed three times for 5 minutes each in a small dish of double distilled water to wash away the mounting medium. The wafer was finally dried at room temperature with a hand blower.”

14) The 'Data analysis' portion of the main text is more focused on data visualization. Some detail is needed regarding the length of traced neurites, etc.

The following has been modified and added in the Results section.

“To demonstrate the suitability of the data for connectomic analysis I traced 9 neurites with starting points located on a 3x3 grid within a central area of the first section, Figure 2C. The neurites spanned the whole depth of the dataset across all sections (10 μm) and exhibited a rather axial orientation, amounting to a total length of at least about 90 μm.”

15) Were any sections distorted during the collection process (e.g. folds, tears, warping), particularly during the mounting step down onto the wafer? Was this assayed?

I added the following paragraphs in the Results and Discussion parts, and a supplementary figure. The discussion about tears has been added in point 4.

Results:

“In dataset 1, 3 tears were found in the EM imagery spanning a total of 9 sections out of the 507 sections. […] Only one tear was present in the magnetic portion of one section (section #22, Figure 2—figure supplement 5B).”

Discussion:

“Quality of the section cutting

A potential concern of MagC is that the iron oxide nanoparticles might damage the fragile diamond edge, thus shortening its lifetime compared to when sectioning a block without magnetic resin. […] As a result, the absence of fold in the collected sections was confirmed in the LM wafer overviews (Figure 1C, Figure 1—figure supplement 1) and when browsing the correlative LM-EM stacks.”

16) Typo subsection “Section collection, last paragraph”, should be 45 mm.

Corrected.

17) The description of 'asymmetrically stacked microscopy coverslips' (subsection “Section collection, last paragraph”) is unclear. A diagram would help to show how the wafer is actually situated in the boat.

Figure 2—figure supplement 5A has been added and referenced from the Materials and methods section.

18) Figure 1—figure supplement 3 references a panel 'H' that is not in the actual figure.

The reference has been removed.

19) Figure 1—figure supplement 1: The sections are of different intensities and contrast. What is the source of this variability?

The following has been added in the caption of Figure 1—figure supplement 1.

“The intensity differences come from section thickness inhomogeneity. As documented by Harris et al., 2006, the lack of an hermetic enclosure during sectioning typically induces section inhomogeneity.”

20) Figure 2—figure supplement 1 references 6 panels, there are only 4 in the figure.

The references have been removed.

21) What is the actual contamination in Figure 1—figure supplement 2, panel D. Is this an aggregation of iron nanoparticles?

I do not have an explanation for the observed contamination, but it looks like an oily contamination. The legend of Figure 1—figure supplement 2) now reads:

**“**D, Small contaminations can sometimes be found in the magnetic resin with an appearance of oily droplet, possibly unreacted oleic acids, an oily compound from handling during preparation, or something unknown to the author.”

22) Figure 1—figure supplement 2, there appear to be slivers detached form the sections in panel a. What are these? I don't see the dummy piece in these sections. Is the attachment of the iron in this example the same as presented in Figure 1?

The slivers are actually exactly the dummy parts. The attachment of the magnetic resin is the same as in Figure 1A. The following paragraph has been added in the legend of Figure 1—figure supplement 2, explaining the mechanism leading to the dummy being attached to the tissue part of the sections. Figure 1—figure supplement 2 has been revised with a yellow box to show one of the dummies in panel A:

“Most of the collected sections exhibit a dummy (one shown in yellow box) that is attached to their tissue part and that comes from the previous section, which is explained by the following mechanism: with this particular sample block, the dummy part often detached from the section during sectioning or immediately after upon landing on the water surface. The main part (magnetic resin and tissue) of the last cut section was then floating freely while its dummy remained attached to the edge. Then the main part of the next cut section sticked to the previous dummy, and so on.”